# Differences in Psychological Variables and the Performance of Female Futsal Players according to Playing Position, Match Result and Menstruation

**DOI:** 10.3390/ijerph20085429

**Published:** 2023-04-07

**Authors:** Trinidad Rico-Lara, Adrián Mateo-Orcajada, Lucía Abenza-Cano, Francisco Esparza-Ros, Raquel Vaquero-Cristóbal

**Affiliations:** 1Facultad de Deporte, UCAM Universidad Católica de Murcia, 30107 Murcia, Spain; 2Cátedra Internacional de Cineantropometría, UCAM Universidad Católica de Murcia, 30107 Murcia, Spain

**Keywords:** activation, female futsal, menstruation, motivation, performance, shots on goal

## Abstract

Futsal performance has been extensively investigated in previous studies, demonstrating that the psychological state of the players or the playing position condition performance. However, research with female indoor soccer players is scarce; therefore, menstruation has not been considered as a determinant aspect for performance in females. Previous research has shown the influence of menstruation on the psychological state and performance of players of different sports; however, there is no known research on female futsal players. For this reason, the aims of the present research were to establish the differences in pre-match psychological variables and offensive performance as a function of playing position, match result and menstruation. A total of 132 s division Spanish players participated in the research. All participants completed the Questionnaire of Psychological Needs of the Athlete-15, and their regular league matches were recorded and subsequently analyzed to determine their offensive performance. The results showed differences according to playing position: pivots and closers showed greater motivation than wings, while the pivots showed greater activation and shots on goal than the wings and closers. Regarding match results, pivots demonstrated more shots on goals than closers only when the match was lost. In addition, the motivation and activation of the pivots, as well as the number of shots on goal, were higher than that of the wings and closers only when they were not menstruating.

## 1. Introduction

Five-side indoor football (futsal) is a sport modality that is currently booming, with more than twelve million players worldwide [1]. Numerous scientific research studies have attempted to analyze its evolution [2] and the factors that influence the players’ performance [3,4]. Thus, previous studies have shown that futsal players require a high level of aerobic endurance as the average match intensity is close to 85–90% of the maximum heart rate. They also require a high anaerobic capacity due to the fact that most of the decisive actions happen in time periods of less than five seconds [5,6]. Among these decisive actions for futsal matches, offensive actions acquire special relevance over defensive actions [7] since they contribute to a greater extent to victory, especially goals, defensive–offensive transitions, shots on goal, assists and precise passes, with these actions being the most frequently studied factors in this area [7,8,9].

However, physical demands are not the only relevant factors in the performance of futsal players since before, during and after the matches there are numerous psychological variables acting as determinants, such as group cohesion, perfectionism, motivation, satisfaction, leadership style or burnout [10]. These variables are interrelated since group cohesion is fundamental in a sport in which interpersonal relationships are established for the achievement of a common goal. However, group cohesion is influenced by variables of an individual nature, such as perfectionism or the motivation of the members of the group [11], which determine the task involvement or well-being of the players on the team [12]. Nevertheless, research in the psychological field on futsal is scarce. Therefore, the relationships between these variables cannot be conclusively established [10].

In addition, it should be noted that the performance in offensive tasks and the psycho-psychological state of futsal players are conditioned by the technical–tactical aspects of the sport, such as the playing position or the outcome of the match [13,14,15,16,17]. In this regard, the tasks performed by the pivots, wings or closers during the games are very different [13]. This gives rise to different profiles, among which stand out the higher motivation, activation, confidence or problem-solving of the pivots with respect to the rest of their teammates [16,17], as well as the increased use of sprinting, jogging and shots on goal [14], allowing them to collaborate in most of the team’s goals [15] and be especially relevant to the final result. In addition, these differences in performance according to playing position are more noticeable when the match result is adverse since there are a high number of counterattacks in which the pivots, as the reference player in the attack, participate to a greater extent in the defensive–offensive transitions [18].

Apart from the influence of the playing position and the outcome of the match, the gender of futsal players is determinant in their performance. Thus, the physical and psychological demands differ considerably between male and female futsal players, with males presenting a better profile for sports performance by showing statistically significant scores in stress control, influence of performance evaluation, motivation, mental ability and team cohesion when compared to females [19]. These differences have a significant impact on the performance and health of futsal players as they influence, among other factors, aspects such as the injury ratio, with previous research conducted in futsal high-level tournaments (Malaysian Games) demonstrating that females had 358.21 injuries per 1000 h of training and competition in the event compared to 247.04 injuries for males, despite the fact that there were no differences in the conditions of competition and general aspects by gender [20]. However, the influence exerted by menstruation on the performance of female futsal players must also be considered [21,22]. In this regard, previous research has shown that certain parameters of sports performance are affected during the menstrual cycle of female elite athletes [22]. In soccer, the decrease in distance run at moderate and high intensities and the number of sprints performed [21], as well as the decrease in attention and affect, were highlighted [23]. In endurance athletes, more than 70% and 50% of the athletes reported reduced fitness and performance, respectively, during the different phases of the menstrual cycle, with bleeding being the phase in which most athletes reported the worst fitness and performance. However, the post-bleeding phase was considered the best for fitness and performance [24]. This is why a large proportion of the athletes reported taking hormonal contraceptives mainly to manipulate the menstrual cycle and prevent the negative effects of the menstrual cycle from interfering with their training and competitions [25]. Despite the extensive previous scientific literature regarding factors that influence the performance of futsal players, previous research conducted in this area with female futsal players is scarce. It is also scarce in the psychological domain [10]. Additionally, there is no known previous scientific research that has analyzed the influence of playing position, match result or menstruation on the performance of female futsal players.

### 1.1. Research Questions

Could the performance and psychological state of female futsal players be influenced by playing position, match result and menstruation?

### 1.2. Aims of Research

The aims of the present research were (a) to establish the differences in pre-match psychological variables and offensive performance as a function of playing position; and (b) to analyze the differences in the study variables as a function of playing position, match result and menstruation.

### 1.3. Hypotheses

Based on previous research in this area and the different demands of each of the positions in futsal, the following research hypotheses are proposed: (H1) there will be significant differences between players of different positions in the psychological state and individual offensive performance variables; and (H2) there will be differences in the pre-match psychological state and individual performance according to menstruation and the outcome of the match.

## 2. Materials and Methods

### 2.1. Design

The design of the present study was descriptive and cross-sectional, with a non-probabilistic sample by convenience with direct and intentional selection of the group. The design and performance of the study were carried out in accordance with the Declaration of Helsinki and the STROBE statement [26]. Prior to the research, the approval of the institutional ethics committee of the Catholic University of Murcia was obtained (code: CE012201). In addition, all participants were informed about the research process and signed an informed consent form prior to the start of the research.

### 2.2. Participants

The sample size was calculated using Rstudio 3.15.0 statistical software (Rstudio Inc., Boston, MA, USA). The standard deviation (SD) from previous research that examined psychological variables in female futsal players (SD = 11.40) was used. [27]. Thus, with an estimated error (d) of 1.99 for a 95% confidence interval, the minimum sample necessary for the development of the research was 125 female players.

The final sample consisted of 132 female futsal players (mean age: 21.90 ± 3.73 years old) belonging to twelve elite teams in the Spanish second division. These athletes performed three training sessions of two hours per week and one match during the 2019/2020 season. It should be noted that this season was affected by the COVID-19 pandemic, with 22 of the 30 games played. All the players participated voluntarily in the present research.

After explaining the aims and the procedure of the research to the trainers of each team, an online informative meeting was held with the players to explain the aims, the questionnaire, the data analysis and the confidential treatment of the data. After this meeting, players who were interested in participating voluntarily provided an informed consent signed by the player and their parents, when the participants were underage.

All female players who participated in the research met the following inclusion criteria: (a) possessed an active federative card from one of the teams in the Spanish second division; (b) did not present any injury that would prevent them from playing matches and/or training sessions; (c) had regular attendance at training sessions and matches (three sessions per week); (d) completed all the questionnaires in their entirety. The exclusion criteria were (a) players who were injured during the match; (b) players who did not participate in the match.

### 2.3. Instruments

The Questionnaire of Psychological Needs of the Athlete-15 (CNPD-15) [28] was used to analyze the psychological state of the player prior to the match. This questionnaire consists of 15 items that assess four dimensions: motivation, activation, confidence and attention. This instrument was previously validated and has a Cronbach’s alpha coefficient of 0.863 for the overall score, as well as 0.672 for motivation (items 1–4), 0.836 for activation (items 5–7), 0.792 for confidence (items 8–11) and 0.809 for attention (items 12–15). For its completion, a Likert scale of 1 to 6 points (1: strongly disagree; 6: strongly agree) was used [28].

Two sociodemographic questions were included in the CNPD-15 questionnaire to determine the player’s playing position in that match and the presence or absence of menstruation.

For the analysis of individual performance during the competition, all regular league matches were recorded using a GoPro8 camera (Hero Black, San Mateo, California, USA), which was placed high enough in the center of the court to cover the entire court. The footage was recorded in 2.7 K and exported at 1080p quality in MP4 format. Subsequently, the performance of each player during the match was analyzed using LongoMatch software v.1.8.3, following the indications of previous research [29,30]. The analysis of the sporting performance of each player in the match with respect to: (a) effective passes, considered the action in which a player passed the ball to another player on her team without having any interference in the trajectory; (b) missed passes, considered a pass from one player to another was that intercepted or stolen by an opposing player or lost on the wing; and (c) shots on goal, considered a shot that was in the direction of the goal and was not intercepted by any opponent, ending in a goal, a shot on the post or a save by the goalkeeper of the opposing team. These actions were selected based on previous research that analyzed the offensive performance in futsal [7,9,31].

### 2.4. Procedure

A protocol was established for data collection that was similar in each match. One hour before the start of each regular league match, the players completed the CNPD-15 questionnaire. The team’s locker room was used for this purpose, in a relaxed atmosphere and without any factors that could interfere with the players’ condition. Once the questionnaire was completed, the players warmed up and played the league match. The camera was placed during the warm-up to ensure that the recording covered the entire court. The entire match was recorded, including halftime.

Once the match was over, the recording was exported and analyzed using Longomatch software v.1.8.3. Two researchers independently analyzed the videos. In cases where there were discrepancies, a third researcher was in charge of establishing agreement. To ensure the validity and reliability of the analysis performed by the researchers, a record sheet was used to determine when each offensive play should be counted. To assess inter-observer reliability, Cohen’s Kappa [32] was used, showing a high level of agreement between the researchers (kappa = 0.904).

The protocol followed during the entire investigation is shown in Figure 1.

### 2.5. Data Analysis

The Kolmogorov–Smirnov test was used to assess the normality of the data, Levene’s test was used to assess the homogeneity and Mauchly’s test was used to assess the sphericity. Since all the variables followed a normal distribution, parametric tests were used for their analysis. Descriptive statistics were used to find the mean values and standard deviation. A one-factor ANCOVA was performed to analyze the differences between pivots, wings and closers, with match result and menstruation as covariates in the model. Subsequently, two MANOVA analyses were performed to establish the differences in the psychological variables and performance variables according to the playing position and the match result and the playing position and menstruation. Bonferroni’s pairwise comparison was used to assess the statistically significant variables. The effect size was calculated using partial eta squared (η^2^), being small (ES ≥ 0.10), moderate (ES ≥ 0.30), large (ES ≥ 1.20) or very large (ES ≥ 2.00), with an error of *p* < 0.05 [33]. A value of *p* < 0.05 was set to determine statistical significance. The statistical analysis was performed using the SPSS statistical package (v.25.0; SPSS Inc., Chicago, IL, USA).

## 3. Results

### 3.1. Pre-Match Psychological Variables and Offensive Performance Differences according to the Playing Position

Table 1 shows the differences in the pre-match psychological variables and the offensive performance variables during the match according to playing position. The differences were significant in motivation (*p* < 0.001) pre-match activation (*p* < 0.001), and shots on goal during the match (*p* < 0.001). The introduction of the covariate match result only showed significant differences in the variable shots on goal (*p* = 0.046), while the covariate presence of menstruation showed significant differences in motivation (*p* = 0.037), activation (*p* = 0.046) and shots on goal (*p* = 0.030).

The analysis of the groups that showed significant differences according to playing position is shown in Table 2. Regarding the psychological variables, the motivation of the pivots (*p* < 0.001) and closers (*p* < 0.001) was higher than the motivation of the wings before the match, while the activation of the pivot was higher than that of the wings (*p* < 0.001) and closers (*p* = 0.037). Regarding the offensive performance variables, the pivots performed a greater number of shots on goal during the match than the wings (*p* < 0.001) and the closers (*p* = 0.014).

### 3.2. Pre-Match Psychological Variables and Offensive Performance Differences according to the Playing Position and Match Result

Table 3 shows the differences in the study variables according to the outcome of the match. Regarding the pre-match psychological variables, the motivation (*p* = 0.002–0.007) and activation (*p* = 0.003–0.018) of the pivots was higher than that of the wings, regardless of the outcome of the match, while in the offensive performance variables, it was found that the pivots showed a greater number of shots on goal than the wings regardless of the outcome (*p* = 0.005–0.048), but only showed more shots on goals than the closers when the match was lost (*p* = 0.049).

### 3.3. Pre-Match Psychological Variables and Offensive Performance Differences according to the Playing Position and Presence of Menstruation

The differences in the study variables according to the presence of menstruation are shown in Figure 2. Regarding the psychological variables, it should be noted that the pre-match motivation of the pivots was higher than the pre-match motivation of the wings when there was no presence of menstruation (*p* < 0.001), while the activation of the pivots was higher than the activation of the wings (*p* = 0.001) and the closers (*p* = 0.019) when there was no presence of menstruation. In the performance variables, the pivots showed a higher number of shots on goal than the wings (*p* < 0.001) and the closers (*p* = 0.001) only when no menstruation was present.

## 4. Discussion

The aims of the present research were to establish the differences in pre-match psychological variables and offensive performance as a function of playing position and to analyze the differences in the study variables as a function of playing position, match result and menstruation. The results of the present study showed that the pre-match motivation of the pivots and closers was higher than the motivation of wings, while the activation of the pivots was higher than that of the wings and closers. It should be noted that, according to the match result, the motivation and activation of the pivots was higher than that of the wings regardless of the outcome of the match. These results are similar to those found in previous research on male soccer players in which forwards were found to be the players with the highest motivation, confidence, activation, problem-solving, stress tolerance, emotional self-awareness, self-regard and impulse control [16,17]. It is important to note that emotional regulation is fundamental for the optimal performance of motor tasks as it is highly related to performance [34,35], with the close relationship between the nature of the task and the degree of arousal being a possible explanation for the greater activation of the pivots. This is because the task of the pivots consists mainly of keeping the ball, disrupting the opponent, accelerating towards the goal and finding the perfect time to score [13], tasks that all require adequate emotional regulation to make decisions and execute actions correctly in a short period of time. Another possible explanation could be that, although it is dependent on the moment of the match and the system of play used, it has been observed that soccer forwards cover the greatest distance in sprinting, repeating this action to a greater extent than the rest of their teammates [36]. This is similar to pivots in futsal, who use sprinting and jogging more frequently [14], entailing a greater energy expenditure that would be related to a greater activation. However, future research in futsal should deepen the relationship between activation and the different parameters of match performance.

Regarding motivation, the results are contrary to those of previous studies conducted on soccer players in which no differences in motivation were found between the different playing positions [37]. This could be due to the fact that the present research was conducted with elite female players, while the previous research was conducted on amateur players [37], since the performance obtained during training and matches is highly related to the extrinsic motivation of the players [38]. Thus, elite players show a higher performance than amateurs in shooting and passing ability, as well as in the ability to sprint [39]. This could be the reason why differences in motivation were found between female pivots and the rest of the players in the present research, since these players perform more shots on goal and sprint more during matches [14]. A high performance in these tasks is especially relevant in the final result, with more than 40% of participation in goals [15].

No differences were found in pre-match confidence or attention according to playing position, even when considering the outcome of the match. Previous research that has analyzed the psychological determinants of performance in futsal female players is scarce [10], but the results previously found are contrary to those of the present research since they showed that the pivots showed lower self-confidence and stress control compared to the rest of the players [27]. A possible explanation for the results found in terms of confidence is that Alvarez-Kurogi’s [27] research was carried out with promising young futsal players under 16 and 19 years of age; this is a stage in which sports performance is associated with the achievement of specific results [40], such as scoring effectiveness in the case of the pivots, and could be a factor to consider since the pivots in the present research were the ones with the most shots on goal compared to the rest of the players. In addition, is important to note that the role played by the coach is fundamental since it is one of the main factors that determines the ego or task orientation of the athletes [40,41]. It is necessary that future research analyzes this orientation in the coach and the players and the possible differences depending on the playing position.

Attending to the offensive performance variables, the pivots took a greater number of shots on goal during the match than the wings, regardless of the outcome of the match. However, it is worth noting that they only took more shots on goal than the closers when the match was lost. These results could be due to the fact that more than 70% of the goals during futsal matches are produced by shots on goal taken from the central offensive zone compared to the defensive and wide zones, an area of the court occupied by the pivots during most of the match [9]. Regarding the match result, the difference between the pivots and closers could be due to the fact that the number of offensive actions is lower during the lost matches and counterattacks are more common, with players in the central areas of the court contributing the most to these actions in quick transitions from defense to attack [18]. However, future research is needed to corroborate these results.

With respect to effective and failed passes, no differences were found when the team won or lost. These results are similar to the results of previous research, showing that successful and unsuccessful passes were not a determining variable for victory or defeat in futsal matches [42]. This could be due to the fact that possession and the number of completed passes do not present enough impact on the final result [31], since previous research conducted in soccer has shown that the significantly determinant actions in offensive performance are assists as they are the last pass that allows a teammate to obtain a positional advantage to successfully finish a play [7]. Due to the scarce number of investigations that have analyzed offensive performance in futsal, future studies should focus on analyzing the type of pass and the area of the court in which it occurs as these factors could be relevant to futsal performance.

An interesting result of the present research was that the pivots showed a higher number of shots on goal than the wings and closers only when they were not menstruating. Previous scientific literature has shown that certain parameters related to sports performance are affected during the menstrual cycle among female elite athletes. However, the results are inconclusive [22]. More specifically, in the field of soccer, it has been observed that the distances covered at moderate and high speeds, the total distance covered, and the number of sprints are significantly different depending on the phase of the menstrual cycle in which the player is, affecting the movement pattern of the players during competitive matches without the total playing time modulating this effect [21]. Thus, the previous research divided the menstrual cycle into three phases: the early follicular phase (menstruation, day 1–4), late follicular phase (day 10–13), and mid-luteal phase (day 20–23). The results showed that the total distance and the distances run at moderate and high intensity were lower during the early follicular phase than in the late follicular phase. In addition, the number of sprints was found to be higher in the late follicular phase than in the early follicular and mid-luteal phases, with pain, discomfort and disturbed mood listed as some of the factors influencing low performance during the early follicular phase [21]. These symptoms seem to be one of the main reasons why sportswomen use hormonal contraceptives, as they allow them to prevent the negative effects of the menstrual cycle and prevent it from interfering with training and competition [25]. This is an aspect to consider since in modalities such as indoor soccer, in which players compete every week, it is more difficult to structure the menstrual cycle for the most decisive competitions, since it depends on the competitive calendar. This is a differentiating factor with respect to men’s futsal, which gives importance to this research. Therefore, the results of the present research are novel because research on performance and menstruation is scarce in the field of futsal. These results provide insight that during menstruation, the performance of the pivots is reduced, probably because futsal is a modality that requires high execution speeds and numerous sprints to achieve performance, and the players’ chances of repeating these types of actions during this phase of the menstrual cycle may be smaller.

Similarly, pivots showed a higher motivation and activation pre-match than wings and closers, but only when they were not menstruating. No previous research is known that has analyzed the influence of the menstrual cycle on the psychological factors of female futsal players. However, in studies conducted with female soccer players, it was observed that attention was significantly lower in the premenstrual phase compared to the follicular and ovulatory phase; this affect was significantly lower during menstruation compared to the ovulatory and luteal phases, while sports performance was significantly higher in the ovulatory phase compared to the menstrual and pre-menstrual phases [23]. As well as increased distraction, more fluctuating emotions and lower motivation were found in female rugby players when they experienced menstruation because they reported great pain and discomfort due to the fact that rugby is a contact sport involving continuous interaction with opponents [43]. The present research shows significant changes in the attention and motivation of the players, with no significant differences found between the pivots and the players of other positions when they were menstruating. This is also consistent with the results of previous research with skiers and biathletes that found changes in mood, symptoms of demotivation and depression, as well as unfocused athletes during the menstruation period [25]. Therefore, the results obtained show that the menstrual cycle seems to influence the performance and psychological factors of futsal players as it occurs in other sports, which is a relevant finding.

The obtained results allow for the acceptance of H1, which indicated that there would be significant differences between players of different positions in the psychological state and individual offensive performance variables since significant differences were found in motivation, activation and shots on goal depending on the playing position. The results also allow for the acceptance of H2, which indicated that there would be differences in the pre-match psychological state and individual performance according to menstruation and the outcome of the match, since significant differences were found in shots on goal between pivots and closers in lost matches. Additionally, higher activation, motivation and shots on goal were also found for pivots compared to wings and closers, yet only when they were not menstruating.

The first novelty of the present study with respect to previous studies is that, there are differences in the pre-game psychological state and in the sport performance during the game, depending on the playing position. In addition, the outcome of the match is relevant for the number of shots on goal. These aspects were previously studied in research on sports modalities, such as soccer, men’s futsal, or rugby, in which no definitive conclusion was obtained since no differences were found in motivation according to playing position in soccer, but the motivation and activation of the pivots was greater than that of the wings in men’s futsal, regardless of the result of the match. However, these aspects have not been analyzed in women’s futsal, a field that lacks scientific research in this regard and has shown that, similar to men’s futsal, performance and psychological variables differ according to playing position and match outcome. But, one aspect must be considered, the physiological differences between male and female athletes, which makes it difficult to compare male and female futsal, since aspects such as menstruation are of great importance in the performance obtained in training and competitions. Thus, the second major novelty that differentiates the present study from previous ones is the inclusion of menstruation as a relevant variable. No previous research has analyzed the influence of menstruation on the performance or psychological state of the futsal players. Therefore, this is a novel aspect, and the results of the present study showed a decrease in the activation and number of shots on goal of the pivots when they were menstruating. This is fundamental since, in addition to the factors that can influence performance in both male’s and female’s futsal, menstruation is a relevant variable in female’s futsal, affecting the psychological state and sporting performance of the players.

The results of the present research can be of great interest for female futsal players and for the technical staff as they show the differences in the performance and the psychological variables of the players according to the playing position. Therefore, it is fundamental to apply specific warm-up protocols in which the motivation and activation needs of each player are considered. These strategies should also be included in the training sessions in an attempt to obtain an adequate level of activation and motivation for each of the players according to their position on the court. Thus, the results would allow coaches and technical staff to optimize the training process by means of psychological interventions. Furthermore, the differences found when considering menstruation provide relevant information for the planning rotations within the teams since it could be a crucial aspect in the performance and psychological state of the players.

The present research is not free of limitations. Since the questionnaire to determine the psychological state of the players was self-completed one hour before the start of the match, and it could be possible that there were small changes in the psychological disposition of the players in the time between the completion of the questionnaire and the start of the match due to the coach’s pre-match talk or the effect of the warm-up on these variables. In addition, it should be considered that there are numerous factors that influence the performance of futsal players: other psychological variables can affect these variables, as can the interaction with teammates or rivals during the match. However, this is one of the first investigations to analyze these factors in women’s futsal, which is a practically unexplored area of scientific literature. This investigation is also the first to introduce menstruation as a variable. Therefore, the present research is a starting point for similar studies on women’s futsal that include other variables that may affect performance. An aspect to consider in future research would be the moment of the menstrual cycle in which the players are, not exclusively the presence or absence of menstruation, since in the present research it has not been taken into consideration and could affect performance This is important, as there is a great deal of evidence in the field of men’s futsal, but the differences between males and females in sport performance are notable, and the results cannot be generalized between sexes [44]. In addition, it would be important for future research to analyze the influence of these factors in first-division women’s futsal teams since the demands are greater physically and psychologically and could condition the players’ performance to a greater extent.

## 5. Conclusions

The results of the present research allow us to conclude that female futsal pivots (*p* < 0.001) and closers (*p* < 0.001) present greater pre-match motivation than wings. Pivots also present greater activation (*p* < 0.001–0.037) and shots on goal (*p* < 0.001–0.014) compared to wings and closers. The result of the match seems to be relevant in the number of shots on goals (*p* = 0.046) mainly when the match is lost, while menstruation seems to influence the psychological (*p* = 0.037–0.046) and performance variables (*p* = 0.030), decreasing the activation and the number of shots on goal of the pivots when they were menstruating. The relevance of these results is high for futsal players and for coaching staff, who can better manage matches and seasons by attending to the psychological demands of the players and their menstrual cycles, factors which are of great relevance for the final performance of the team.

## Figures and Tables

**Figure 1 ijerph-20-05429-f001:**
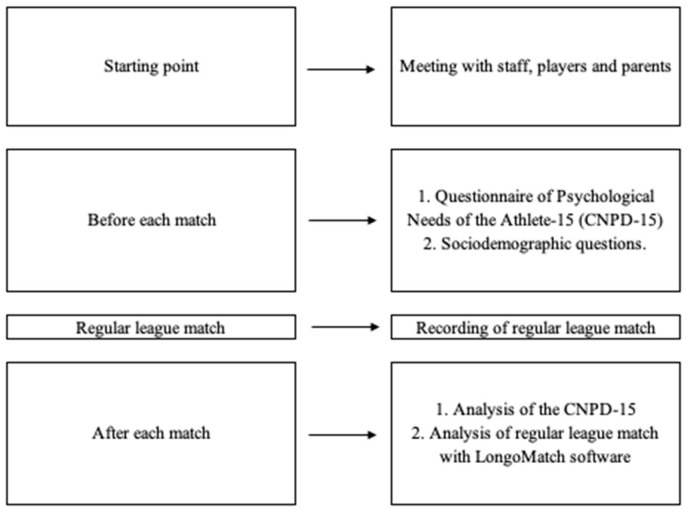
Protocol established for the investigation.

**Figure 2 ijerph-20-05429-f002:**
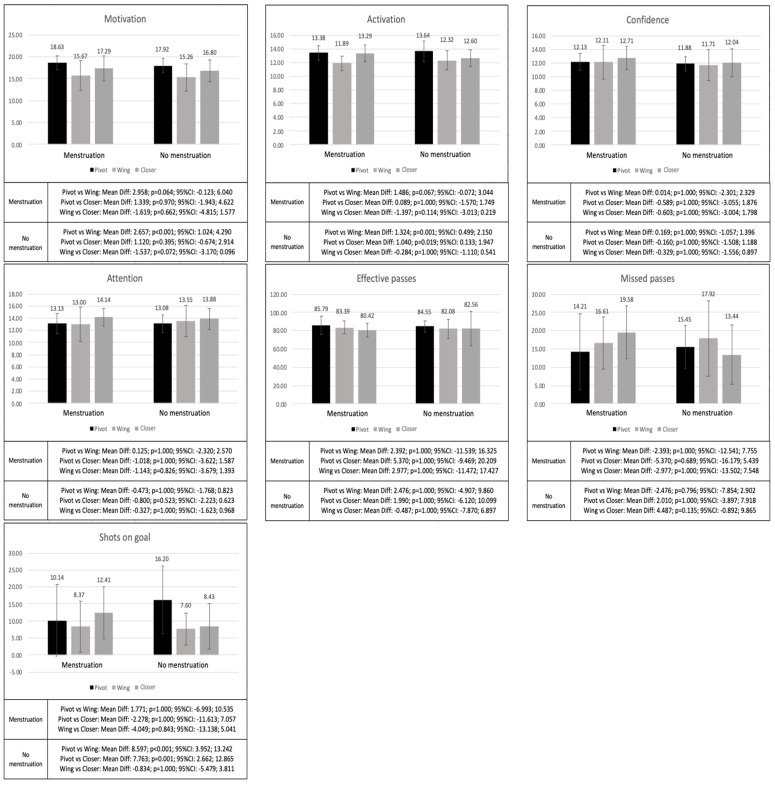
Differences in the physiological and offensive performance variables between the players with different playing position according to the presence of menstruation.

**Table 1 ijerph-20-05429-t001:** Differences in the physiological and offensive performance variables between the players with different playing positions and the influence of match result and menstruation.

Variable	Mean ± SD			F	*p*	Effect Size	Position × Match Result			Position × Menstruation		
	Pivots (n = 33)	Wings (n = 47)	Closers (n = 32)				F	*p*	Effect Size	F	*p*	Effect Size
Motivation	18.09 ± 1.63	15.34 ± 3.12	16.91 ± 2.49	11.329	<0.001	0.172	0.283	0.754	0.005	0.023	0.037	0.081
Activation	13.58 ± 1.39	12.23 ± 1.30	12.75 ± 1.24	10.100	<0.001	0.156	0.008	0.992	0.001	1.230	0.046	0.073
Confidence	11.94 ± 1.08	11.79 ± 2.32	12.19 ± 2.00	0.405	0.668	0.007	0.247	0.782	0.005	0.070	0.932	0.001
Attention	13.09 ± 1.46	13.45 ± 2.58	13.94 ± 1.60	1.403	0.250	0.025	0.256	0.775	0.005	0.272	0.762	0.005
Effective passes	84.92 ± 6.94	82.35 ± 9.61	82.10 ± 1.89	0.643	0.528	0.011	0.597	0.552	0.011	0.164	0.849	0.003
Missed passes	15.08 ± 6.94	17.65 ± 9.61	14.78 ± 8.26	1.421	0.246	0.024	0.386	0.681	0.007	1.451	0.239	0.027
Shots on goal	14.84 ± 10.12	7.58 ± 5.34	9.30 ± 6.93	9.695	<0.001	0.138	0.045	0.046	0.071	2.809	0.030	0.100

**Table 2 ijerph-20-05429-t002:** Bonferroni post hoc of the differences in the physiological and offensive performance variables between the players with different playing positions.

Variable	Group 1	Group 2	Mean Diff. G1–G2	*p*	95% CI
Motivation	Pivot	Wing	2.760	<0.001	1.328; 4.191
	Pivot	Closer	1.182	0.206	−0.381; 2.745
	Wing	Closer	−1.578	<0.001	−3.023; −0.133
Activation	Pivot	Wing	1.331	<0.001	0.604; 2.057
	Pivot	Closer	0.829	0.037	0.036; 1.622
	Wing	Closer	−0.501	0.298	−1.235; 0.232
Confidence	Pivot	Wing	0.151	1.000	−0.927; 1.228
	Pivot	Closer	−0.248	1.000	−1.424; 0.929
	Wing	Closer	−0.398	1.000	−1.486; 0.690
Attention	Pivot	Wing	−0.351	1.000	−1.486; 0.785
	Pivot	Closer	−0.848	0.297	−2.088; 0.391
	Wing	Closer	−0.498	0.880	−1.644; 0.649
Effective passes	Pivot	Wing	2.586	0.995	−3.864; 9.036
	Pivot	Closer	2.738	1.000	−4.304; 9.780
	Wing	Closer	0.153	1.000	−6.359; 6.664
Missed passes	Pivot	Wing	−2.500	0.615	−7.266; 2.266
	Pivot	Closer	0.359	1.000	−4.845; 5.563
	Wing	Closer	2.859	0.454	−1.953; 7.671
Shots on goal	Pivot	Wing	6.973	<0.001	2.806; 11.140
	Pivot	Closer	5.426	0.014	0.877; 9.975
	Wing	Closer	−1.547	1.000	−5.754; 2.660

**Table 3 ijerph-20-05429-t003:** Differences in the physiological and offensive performance variables between the players with different playing position according to the match result.

Variable	Match Result	Position	Mean ± SD	Compared with	Mean Diff.	*p*	95% CI
Motivation	Win	Pivot	18.29 ± 1.326	Wing	3.286	0.002	1.025; 5.546
		Wing	15.00 ± 2.970	Closer	−1.714	0.204	−3.975; 0.546
		Closer	16.71 ± 2.431	Pivot	−1.571	0.341	−3.969; 0.826
	Lost	Pivot	17.95 ± 1.840	Wing	2.396	0.007	0.523; 4.268
		Wing	15.55 ± 3.247	Closer	−1.504	0.172	−3.407; 0.400
		Closer	17.06 ± 2.600	Pivot	−0.892	0.902	−2.978; 1.195
Activation	Win	Pivot	13.71 ± 1.647	Wing	1.325	0.018	0.176; 2.475
		Wing	12.39 ± 1.335	Closer	−0.540	0.768	−1.690; 0.610
		Closer	12.93 ± 1.207	Pivot	−0.786	0.360	−2.005; 0.434
	Lost	Pivot	13.47 ± 1.219	Wing	1.336	0.003	0.383; 2.288
		Wing	12.14 ± 1.302	Closer	−0.473	0.712	−1.441; 0.495
		Closer	12.61 ± 1.290	Pivot	−0.863	0.152	−1.924; 0.199
Confidence	Win	Pivot	11.79 ± 1.122	Wing	−0.214	1.000	−1.916; 1.488
		Wing	12.00 ± 2.521	Closer	−0.143	1.000	−1.845; 1.559
		Closer	12.14 ± 1.994	Pivot	0.357	1.000	−1.448; 2.162
	Lost	Pivot	12.05 ± 1.079	Wing	0.397	1.000	−1.012; 1.807
		Wing	11.66 ± 2.224	Closer	−0.567	1.000	−2.000; 0.866
		Closer	12.22 ± 2.074	Pivot	0.170	1.000	−1.401; 1.741
Attention	Win	Pivot	13.21 ± 1.311	Wing	−0.175	1.000	−1.968; 1.619
		Wing	13.39 ± 2.993	Closer	−0.254	1.000	−2.047; 1.539
		Closer	13.64 ± 1.447	Pivot	0.429	1.000	−1.473; 2.331
	Lost	Pivot	13.00 ± 1.599	Wing	−0.483	1.000	−1.968; 1.002
		Wing	13.48 ± 2.355	Closer	−0.684	0.819	−2.194; 0.826
		Closer	14.17 ± 1.724	Pivot	1.167	0.268	−0.488; 2.822
Effective passes	Win	Pivot	85.65 ± 7.20	Wing	4.028	1.000	−6.127; 14.183
		Wing	81.62 ± 11.49	Closer	2.329	1.000	−7.826; 12.484
		Closer	79.29 ± 24.26	Pivot	−6.357	0.462	−17.128; 4.414
	Lost	Pivot	84.27 ± 7.05	Wing	1.497	1.000	−6.914; 9.908
		Wing	82.77 ± 8.62	Closer	−1.505	1.000	−10.056; 7.046
		Closer	84.28 ± 7.68	Pivot	0.008	1.000	−9.365; 9.381
Missed passes	Win	Pivot	14.35 ± 7.20	Wing	−4.028	0.586	−11.547; 3.491
		Wing	18.38 ± 11.49	Closer	4.814	0.367	−2.705; 12.333
		Closer	13.57 ± 9.11	Pivot	−0.786	1.000	−8.761; 7.189
	Lost	Pivot	15.73 ± 7.05	Wing	−1.497	1.000	−7.725; 4.731
		Wing	17.23 ± 8.62	Closer	1.505	1.000	−4.826; 7.836
		Closer	15.73 ± 7.68	Pivot	−0.008	1.000	−6.948; 6.932
Shots on goal	Win	Pivot	14.51 ± 11.80	Wing	6.630	0.048	0.036; 13.225
		Wing	7.88 ± 6.42	Closer	−1.858	1.000	−8.452; 4.736
		Closer	9.74 ± 6.94	Pivot	−4.772	0.300	−11.767; 2.222
	Lost	Pivot	14.89 ± 9.30	Wing	7.224	0.005	1.762; 12.686
		Wing	7.66 ± 4.56	Closer	−1.304	1.000	−6.857; 4.249
		Closer	8.97 ± 7.11	Pivot	−5.921	0.049	−12.008; 0.166

## Data Availability

The data presented in this study are available on request from the corresponding author. The data are not publicly available due to containing information that could compromise the privacy of research participants, but data are available from the corresponding author on reasonable request.

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
