# Peer review of "Differences in Psychological Variables and the Performance of Female Futsal Players according to Playing Position, Match Result and Menstruation"

_ijerph, 2023, doi:10.3390/ijerph20085429_

Round 1

Reviewer 1 Report

The introduction properly introduces the problem of research, it not only characterizes the game in futsal, but also attempts to analyze the requirements of this game for the differentiation of men and women, in whom the efficiency of the game may be disturbed by the menstrual cycle. This problem is important because in the professional literature on the subject these aspects are little recognized This part of the work contains the aim, research questions and hypotheses. They are formulated correctly, however, according to the reviewer, for full transparency of the study, they should be in a separate section entitled: Aim of research, research questions and hypotheses. According to the reviewer, the research material and methods meet high requirements in terms of accuracy and reliability. The research results are precisely presented in accordance with the purpose of the research. Also, the discussion refers to the research results and responds to the research hypotheses, it is conducted with the logic of the data obtained with reference to the literature of the subject. According to the reviewer, research results (tables) and diagrams should no longer be presented in the discussion. This should be in the Research Results section. According to the reviewer, such a picture of the study is inconsistent with the methodology of the work. I leave this fact to the editor's decision

Author Response

Reviewer 1

The introduction properly introduces the problem of research, it not only characterizes the game in futsal, but also attempts to analyze the requirements of this game for the differentiation of men and women, in whom the efficiency of the game may be disturbed by the menstrual cycle. This problem is important because in the professional literature on the subject these aspects are little recognized. 

+ Dear reviewer, thank you very much for your review of our manuscript. We have addressed the changes you indicate below to improve the quality of the manuscript.

This part of the work contains the aim, research questions and hypotheses. They are formulated correctly, however, according to the reviewer, for full transparency of the study, they should be in a separate section entitled: Aim of research, research questions and hypotheses.

+ Thank you very much for your contribution. Three subsections have been included at the end of the introduction: 1.1 Research questions; 1.2 Aims of research; 1.3 Hypotheses.

According to the reviewer, the research material and methods meet high requirements in terms of accuracy and reliability. The research results are precisely presented in accordance with the purpose of the research. 

+ Thank you very much.

Also, the discussion refers to the research results and responds to the research hypotheses, it is conducted with the logic of the data obtained with reference to the literature of the subject. 

+ Thank you very much for your comment.

According to the reviewer, research results (tables) and diagrams should no longer be presented in the discussion. This should be in the Research Results section. 

+ Thank you very much for your contribution. The tables and figures have been moved to belong to the results section.

According to the reviewer, such a picture of the study is inconsistent with the methodology of the work. I leave this fact to the editor's decision.

+ Thank you very much for your contributions. We have tried to attend to all of them and we believe that the quality of the manuscript has improved considerably. We remain at your disposal for any further information you may need.

Reviewer 2 Report

Dear authors

Thank you for your article

I have put my comments in the text directly

Author Response

Reviewer 2

Line 65: on what criteria do you say that men have a better psychological state than women?

+ Dear reviewer, thank you very much for your input. We have specified the specific variables in which boys show higher scores and therefore have a better profile to perform in futsal.

Line 68: for the same level of play and the same number of training hours?

+ Thank you very much for your comment. The tournament and conditions (being similar for male and female) have been specified.

Line 105: for the 2019-2020 season, these data must have been only partial given the arrival of the covid pandemic from February 2020, please clarify this.

+ Thank you very much for your great contribution. It has been specified that the season did not end, and data was only collected for 22 of the 30 matches.

Line 106: What is the level of training in the Spanish women league 2, is it the same as in the Spanish league 1 and why didn’t you also ask the Spanish women league 1?

+ Thank you very much for your contribution. The frequency and duration of team training has been specified. It was not possible to access the first division teams because they have greater demands (they train 4-5 times a week and travel longer distances throughout Spain), which made it impossible to collect this sample. This has been included in future lines of research.

Table 1: please change the position of the table as it is cut in two.

+ Thank you. Done.

Line 343: please specify when in the menstrual cycle and whether or not the periods were painful and whether or not they actually affected the women in their practice.

+ Thank you very much for your interesting contribution. The requested information has been included.

Line 354: please be more specific about the impact of menstruation on the psychological aspect of women during their physical activity.

+ Thank you very much. It has been specified how psychological variables vary depending on menstruation.

+ Thank you very much for your contributions. We have tried to attend to all of them and we believe that the quality of the manuscript has improved considerably.

Reviewer 3 Report

·         The authors need to change the abstract and focus more on problem domain. Before developing the proposed method.

·         The novelty of this paper is not clear. The difference between present work and previous Works should be highlighted.

·         The authors could better explain how “Related works” is actually related to the current study. It is not clear to the reader how the manuscript is similar to or differs from these related works.

·         Results need more explanations. Additional analysis is required at each experiment to show the main purpose.

·         Authors must develop the framework/architecture of the proposed methods

·         The authors have used some mathematical notations. Make sure that all the parameters are described. And also check the mathematical notations.

·         How did the authors apply the Augmentation technique?

·         The manuscript is well-organized and properly formatted. The authors are suggested to have the paper revised to improve the language.

·          The conclusion part should indicate the implications of the experimental evaluation and include some obtained values to point out the superiority clearly.

Author Response

Reviewer 3

-  The authors need to change the abstract and focus more on problem domain. Before developing the proposed method.

+ Thank you very much for your contribution. More background that contextualizes the research and gives importance to the research variables has been included before the method.

- The novelty of this paper is not clear. The difference between present work and previous Works should be highlighted.

+ Thank you very much for your contribution. A paragraph has been included reporting the main novelties of the present research with respect to previous studies.

- The authors could better explain how “Related works” is actually related to the current study. It is not clear to the reader how the manuscript is similar to or differs from these related works.

+ Thank you very much. A comment has been included in the final part of the discussion in which the differences of the article with respect to previous research are discussed.

- Results need more explanations. Additional analysis is required at each experiment to show the main purpose.

+ Thank you for this comment. Additional explanations have been added to the results section to improve understanding. If you have any other specific suggestions, please let us know.

- Authors must develop the framework/architecture of the proposed methods

+ Thank you very much. A new Figure (Figure 1) has been included to explain the proposed methods.

- The authors have used some mathematical notations. Make sure that all the parameters are described. And also check the mathematical notations.

+ Dear reviewer, thank you very much for your comment. We have carefully checked the mathematical notations to avoid errors in them.

- How did the authors apply the Augmentation technique?

+ Thank you very much. We only used the scaling method to increase the size of the graphics so that they could be viewed correctly. If this is not what you are referring to, please let us know.

- The manuscript is well-organized and properly formatted. The authors are suggested to have the paper revised to improve the language.

+ Thank you very much. The article has been sent to an expert translator to make the necessary modifications.

- The conclusion part should indicate the implications of the experimental evaluation and include some obtained values to point out the superiority clearly.

+ Thank you very much for your contribution. The requested aspects have been included to improve the conclusion section.

+ Thank you very much for your contributions. We have tried to attend to all of them and we believe that the quality of the manuscript has improved considerably.
